# LiDAR-Based GNSS Denied Localization for Autonomous Racing Cars

**DOI:** 10.3390/s20143992

**Published:** 2020-07-17

**Authors:** Federico Massa, Luca Bonamini, Alessandro Settimi, Lucia Pallottino, Danilo Caporale

**Affiliations:** 1Research Centre E. Piaggio, Università di Pisa, 56122 Pisa, Italy; lucabonamini28@gmail.com (L.B.); ale.settimi@gmail.com (A.S.); lucia.pallottino@unipi.it (L.P.); d.caporale@centropiaggio.unipi.it (D.C.); 2Dipartimento di Ingegneria dell’Informazione, Università di Pisa, 56122 Pisa, Italy

**Keywords:** LiDAR signal processing, sensor and information fusion, advanced driver assistance systems, autonomous racing

## Abstract

Self driving vehicles promise to bring one of the greatest technological and social revolutions of the next decade for their potential to drastically change human mobility and goods transportation, in particular regarding efficiency and safety. Autonomous racing provides very similar technological issues while allowing for more extreme conditions in a safe human environment. While the software stack driving the racing car consists of several modules, in this paper we focus on the localization problem, which provides as output the estimated pose of the vehicle needed by the planning and control modules. When driving near the friction limits, localization accuracy is critical as small errors can induce large errors in control due to the nonlinearities of the vehicle’s dynamic model. In this paper, we present a localization architecture for a racing car that does not rely on Global Navigation Satellite Systems (GNSS). It consists of two multi-rate Extended Kalman Filters and an extension of a state-of-the-art laser-based Monte Carlo localization approach that exploits some a priori knowledge of the environment and context. We first compare the proposed method with a solution based on a widely employed state-of-the-art implementation, outlining its strengths and limitations within our experimental scenario. The architecture is then tested both in simulation and experimentally on a full-scale autonomous electric racing car during an event of Roborace Season Alpha. The results show its robustness in avoiding the robot kidnapping problem typical of particle filters localization methods, while providing a smooth and high rate pose estimate. The pose error distribution depends on the car velocity, and spans on average from 0.1 m (at 60 km/h) to 1.48 m (at 200 km/h) laterally and from 1.9 m (at 100 km/h) to 4.92 m (at 200 km/h) longitudinally.

## 1. Introduction

Aside from the classical uses of automated human transportation in urban scenarios, another  exciting application of self-driving technologies comes with autonomous racing, intended as the competition between a self-driving race car and other vehicles, either human driven or autonomous. This scenario is particularly interesting as it represents an extreme yet safe condition in which to test autonomous driving algorithms: extreme because of high speeds and accelerations at the limit of friction, safe as the vehicle runs the artificial driver software in a racing track and does not transport humans. It is worth noticing that many of the problems faced in autonomous racing are the same as those found in more common scenarios, such as urban or highway autonomous driving, in particular regarding self state estimation tasks such as localization, which is the focus of this paper.

Any self-driving vehicle relies on localization systems to provide an accurate pose estimation to efficiently and safely navigate in the environment. Such estimates are obtained by means of sensor fusion algorithms that combine information coming from different sources, possibly at different rates. The key factors in evaluating these methods are accuracy, responsiveness, robustness and reliability in the presence of signal degradation [1,2].

In autonomous racing, these factors are inherently critical, as to achieve optimal behavior, two main problems arise:While facing a tight turn or passing through narrow corridors, the optimal trajectory is close to the internal edge of the turn or, more in general, the track border;The optimal speed profile is at the limit of friction, thus, small localization errors can lead to divergent behaviors.

An extensive literature exists on the more general problem of simultaneous localization and mapping (SLAM), where there are basically two approaches based on filtering or optimization methods, see the following surveys [2,3,4,5,6]. In [7,8], the authors proposed an efficient solution to implement Rao–Blackwellized particle filters to obtain occupancy grid maps of the environments. In [9], the use of probabilistic maps is extended for dynamic urban environments. In [10], the authors developed a complete navigation stack for the Defense Advanced Research Projects Agency (DARPA) Challenge using 3D Light Detection and Ranging (LiDAR), Global Positioning System (GPS) and inertial sensors for the localization task. In [11], an efficient (in terms of data storage) localization method based on 3D LiDARs without the use of GPS is described. In [12,13], LiDAR-based systems with combined GNSS data are described. In [14], the authors show that it is preferrable to have a LiDAR mounted on top of the vehicle with a 360 degrees field of view than multiple planar LiDARs mounted around the vehicle, as is the vehicle considered in this work, both for ease of operation and localization performance. Similar conclusions are drawn in [15], where the authors compare the performance of a localization system with an INS (Inertial Navigation System) and camera, with or without 2D/3D LiDAR, highlighting the important contribution of LiDAR sensors for this task. A survey on different localization methods for autonomous vehicles can be found in [16]. As for SLAM, two main approaches are used in autonomous driving: optimization-based and filter-based ones. In this work we leverage on a state-of-the-art implementation of Adaptive Monte Carlo Localization (AMCL) [17]. A common pitfall of approaches based on particle filters is the so-called kidnapping problem, which becomes particularly dangerous during a race. Deep-learning methods are generally used in camera-based systems, for instance in [18] for visual SLAM based on monocular camera input. Recently a deep-learning classifier has been proposed in [19] to detect kidnapping in particle filter based localization, and a Long-Short Term Memory network has been used in [20] to quickly recover from kidnapping of a mobile robot. It has been recently shown in [21] that a deep-learning and Monte Carlo Localization method can reduce localization time even in large maps. In that work, a training set of 35 h and over 150 km, which is still too far from our use case where generally less than 10 h are available and collected in the same week of the race. More generally, online diagnosis of failure of perception systems for autonomous driving is an open problem, and a recent mathematical framework has been proposed in [22]. Optimization-based approaches (see for example [23]) tend to suffer less from the kidnapping problem, at least in the authors’ experience, but, on the other hand, are typically computationally intensive. A solution to achieve real time SLAM on Intel CPU using 3D LiDAR point clouds from a Velodyne sensor has been proposed in [24]. In our experience, both approaches are challenged in race tracks by the presence of straights. This is a problem especially when the LiDAR scans are matched to the map with the Iterative Closest Point (ICP) method [25]. An interesting solution was proposed in [26] in the context of train localization in tunnels by means of Rényi quadratic entropy. Normal Distribution Transform methods have been proposed in [27] for 3D mapping of the environment, and are promising for applications with 3D LiDARs even for self-driving applications.

The sensor setup is the first aspect to consider when designing a state estimation system. In a typical autonomous car, available sensors are optical speed sensors, wheel encoders, Inertial Measurement Unit (IMU) (propioceptive, working at frequencies of hundreds of Hz), LiDARs and cameras (exteroceptive, working at frequencies of tens of Hz). In order to achieve the required steering and traction control accuracy, it is necessary that the state estimation module outputs a high frequency signal (hundreds of Hz) with sufficient accuracy [28,29]. This multi-rate environment calls for specific solutions for state estimation that we address in this paper. Furthermore, as the racing context can be arbitrarily well structured, depending on the competition rules (pre-known track map, controlled weather, known type of track edges, known features of interacting agents), there is a lot of a priori information that can be exploited to enhance the quality of the state estimation [30,31,32].

Referring to GNSS, which can provide high frequency global pose estimates, is not always a viable solution. Despite being very reliable in open-sky environments, it can quickly stop being so in the presence of multiple sources of signal degradation [3,33]. Even in urban racing tracks such as those of Formula E (https://www.fiaformulae.com/) events, GNSS is affected by the presence of nearby buildings, trees, or tunnels. Referring to Real-Time Kinematic (RTK) antennas can mitigate this problem; however, the degradation can still be unacceptable in obstacle-dense scenarios and requires a widespread infrastructure. This is not a problem limited to autonomous cars, but it is very common for indoor navigation tasks. In [34,35], for example, the authors tackle the problem of navigation of a micro air vehicle in a GNSS-denied environment.

This motivated our choice of developing a system capable of not relying at all on any signal coming from GNSS for localization.

With respect to our previous works, where an optimization based approach was used [36,37], we adopted a method based on particle filtering due to the lower computational burden required on the specific hardware and, in perspective, because it is more amenable to parallelization than optimization based approaches. The aforementioned drawback of this method (the kidnapping problem) was solved by injecting a priori knowledge of the particular scenario into the filter, as will be explained in detail in Section 5, thus representing a valid alternative to optimization-based algorithms.

The testbed for the proposed architecture was the first Roborace (https://roborace.com/) Localization Challenge, which was created with the same goal in mind, which is to prove the capability of racing with degraded GNSS reception. To the best of our knowledge, this is the first autonomous racing challenge of this kind, an important step towards the realization of a more realistic racing challenge.

The race rules are simplified, consisting of a race with a single vehicle and no other moving obstacles on the car path, but still pose some interesting challenges as the car (2 m wide) was required to pass through several narrow gates (2.5 m wide) while driving at the maximum speed allowed by the track. A maximum speed of 100 km/h was imposed given the characteristics of the track, which is shown in Figure 1.

In this paper we report on the whole state estimation system, consisting of two multi-rate Kalman Filters for odometry estimation and smoothing, a Madgwick Filter for orientation estimation and a LiDAR processing algorithm that we named Informed Adaptive Monte Carlo localization (IAMCL), which was tested in simulation and on a real racing vehicle in the context of the localization challenge.

As the name suggests, this algorithm is an extension of the classical Monte Carlo localization algorithm and it is based on the famous AMCL implementation within the Robot Operating System (ROS)  [17], where some a priori knowledge on the scenario is injected to enhance performances and prevent the kidnapping problem.

In the following, before presenting the proposed localization method in detail, we outline the problem formulation, the underlying sensors setup, and the mapping procedure. Finally, we report on simulation and experimental results.

## 2. Problem Formulation

We focus on the development of a localization system relying on 2D LiDAR input in the absence of GNSS signal. We assume that an occupancy grid map is built before localization. We also assume that the LiDAR point cloud model is known and that, during localization, the occupancy grid map is noiseless.

This assumption is removed while generating the map, a process detailed in Section 4. More specifically, our goal is to provide a high frequency and smooth pose estimate for a car moving in a race track. To achieve this, we leverage upon a widely used implementation in the mobile robotics community (AMCL package in ROS) extending it to avoid known issues that hinder its usability in a racing setting: the need for manual initialization, which is prone to human error and is in general not efficient, and the kidnapped robot problem, which constitutes a safety concern for a racing car due to the possibility of sudden discontinuities in the pose estimate and the resulting feedback control actions.

Let the kinematic state of the vehicle be q=(x,y,φ,u), for which we need to fuse a set of asynchronous proprioceptive sensor measurements with a LiDAR point cloud. We do not include, during localization, direct measurements of the vehicle pose (x,y,φ), as no GNSS/magnetometer data is available. The frame of reference for *q* is taken within a pre-built occupancy grid map that is also used for LiDAR scan matching. The vehicle control system requires a 250 Hz pose estimate [36]: while the on board velocity and acceleration sensors are able to provide signals at this (or higher) frequency, the LiDAR scans are provided at 25 Hz. Hence, a smooth pose estimate at high frequency has to be carefully computed from sources with multiple rates.

## 3. Sensors Setup

Roborace’s DevBot 2.0 (London, UK), the car used during the race, has several sensors available, a subset of which is relevant to this work shown in Figure 2. For an overview of the vehicle’s hardware architecture, see our previous work [36]. For the sake of this paper, the sensor data comes from the following three sources:**OxTS Inertial Navigation System (INS):** this commercial module consists of a dual-antenna GNSS and an IMU , which are pre-fused to obtain a high frequency (250 Hz) pose, velocity, and accelerations estimates;**Ibeo LiDAR range finders**: four LiDAR sensors are placed on the corners of the vehicle, each with 4 vertically stacked layers at 0.8 degrees spacing. The aggregate point cloud resulting from all the sensors is provided with a rate of 25 Hz.**Optical Speed Sensor (OSS):** this sensor provides direct longitudinal and lateral speed measurements through a Controller Area Network (CAN) interface at 500 Hz, and it is not affected by wheel drift.

Finally, an RTK base station is placed at a fixed position near the track, which can significantly improve the position accuracy provided by GNSS. Note that, when this system is on, the GNSS accuracy is so high (in our experimental context) that it solves the localization problem, as the measurement has an error of the order of a few millimeters. Thus, we consider the data coming from this system as a truth reference to compute the localization error metrics, while during the actual race this system is deactivated and the number of satellites is limited to simulate a GNSS-denied scenario.

We consider the reference frame depicted in Figure 3 with the *x*-axis of the velocity laying along the forward direction, and the *y*-axis on the left. The INS system provides also absolute position estimates with respect to the map origin and orientation φ relative to the cardinal east, longitudinal and lateral speeds *u* and *v*, longitudinal and lateral accelerations $a_{x}$ and $a_{y}$, and angular velocity along the yaw axis ω.

Note that there are in theory two sensors that can provide vehicle speed measurements: INS and OSS. We decided to rely on OSS due to the fact that, while INS uses GNSS inputs to provide these measurements, thus producing unreliable estimates when that information is denied, OSS is totally independent from it. Thus, we rely on INS only for accelerations and angular rates. This way we obtained a more realistic simulation of a scenario with complete GNSS absence.

An estimate of the orientation φ is instead obtained from a Madgwick filter that fuse angular velocities and accelerations from the INS system, as will be described in Section 5.

## 4. Mapping Procedure

Mapping of the track is performed in dedicated sessions where the car is manually driven within the track borders for data acquisition and offline map generation. During these sessions it is permitted to use RTK-GNSS data that, being extremely accurate, make localization not an issue. Finally, the map is generated with OpenSlam’s Gmapping [7], a particle filter based algorithm that builds grid maps through the use of LiDAR and odometry data. Note that, although localization is assumed accurate, the resulting map can still feature some distortions due to the point cloud noise, the LiDAR calibration error and the algorithm parameters tuning. In Figure 4 the resulting map of the racing track (Zala Zone circuit) is shown. In the following, we outline a procedure to qualitatively assess the accuracy of the map used while tuning the mapping algorithm.

### Map Quality Assessment

As described in Section 3, for localization we only rely on velocities, accelerations and on the LiDAR data. Of these, only LiDAR data provide an absolute pose estimate for the vehicle, which in our approach is obtained by matching the data with the offline built map, as will be described in Section 5. Thus, the localization performance is inherently limited by the map quality.

Issues such as distortions in the map result in a bias *b* of the final pose estimate that is not easily removable, thus we have:$$E\left\{\hat{\xi }\left(s\right)\right\}=E\left\{\xi^{*}\left(s\right)\right\}+b\left(s\right)$$
where E is the statistical expectation operator, $\hat{\xi }$ is the car estimated pose, $\xi^{*}$ is the car true pose, and *s* is the arc length distance along the racing track. This means that the empirical localization error with respect to GNSS consists of two contributions, the first being the actual localization error, and the second being influenced by the mapping error, which is not only unknown but also variable along the track.

The bias is qualitatively evaluated by measuring how well the LiDAR data matches the map when fixing the vehicle’s pose at the GNSS/RTK position. To do this, we consider the average mapping error:$$e_{\mathrm{map}}\left(\xi^{*}\left(k\right)\right)=\frac{\sum_{i=1}^{\Lambda }|T_{\xi^{*}\left(k\right)}\left[p_{i}\left(k\right)\right]-m\left(T_{\xi^{*}\left(k\right)}\left[p_{i}\left(k\right)\right])\right)|}{\Lambda },$$
where *k* is the time instant, Λ is the number of laser scan points, $\xi^{*}$ is the value of the GNSS/RTK pose estimate, $p_{i}$ is the coordinate of a particular laser scan point, $T_{q}\left[\cdot \right]$ is a roto-translation defined by the pose *q*, and m(·) is a function that takes a laser scan point as input and returns the coordinate of the closest occupied point in the occupancy grid map. The procedure for a single given GNSS pose is illustrated in Figure 5a.

This procedure is iterated along the map, and its results are shown in Figure 5b, where colors ranging from green to red indicate a progressively worse value of the average mapping error. The result is compatible with the idea that mapping is more accurate near distinctive features of the track, i.e., elbows, whereas it shows worse results on straights, likely because of the LiDAR sensor giving mostly planar information. The proposed procedure gives an intuitive representation of the mapping quality, and it is used to evaluate different mappings and to tune mapping algorithms.

## 5. State Estimation Algorithm

An overview of the proposed localization system is reported in Figure 6. Its main components are an Extended Kalman Filter (EKF1) used to compute a high frequency velocity estimate, a LiDAR-based particle filter called IAMCL used to compute a low frequency vehicle pose estimate by comparing the LiDAR data with the offline built map, and a final Extended Kalman Filter (EKF2) used to provide a smooth, high frequency and accurate pose estimate to the vehicle control system.

Some design choices were constrained by practical considerations related to the hardware available on the Roborace DevBot, specifically an NVIDIA DRIVE PX2 board with a non real-time Linux Kernel [36]. Thus, important constraints were in place in terms of:Computational burden: the NVIDIA Drive PX2 has a high number of CUDA cores, while CPU is rather limited; in this paper we propose a CPU implementation for simplicity and because the available computing power was enough for the particular experimental task;Flexibility: the particular race format affects not only strategy but also which sensors are available and what other modules must concurrently run on the board (e.g., interfaces with V2X race control infrastructure, planning software);Real-time requirements: the DevBot motion control module runs in real-time on a dedicated SpeedGoat board at 250 Hz. No patch was allowed to the standard Ubuntu kernel to make it real-time compliant. Thus, it needs to receive a pose estimate signal with high frequency.

In the following we describe the architecture of the three filters.

### 5.1. Odometry (EKF1)

This filter provides an odometry pose estimation. The goal of this filter is to provide the next filter (see Section 5.2) with a good velocity and orientation estimate. This filter consists of a Madgwick filter [38] for the orientation estimation and of an Extended Kalman Filter based on 4D state dynamics, in which the evolution follows a simple unicycle model (as in [39]):(1)$$\begin{array}{cc} x_{k+1} & =x_{k}+u_{k}\cos\left(\varphi_{k}\right)\Delta t\\ y_{k+1} & =y_{k}+u_{k}\sin\left(\varphi_{k}\right)\Delta t\\ \varphi_{k+1} & =\varphi_{k}+{\hat{\omega }}_{k}\Delta t\\ u_{k+1} & =u_{k}+{\hat{a}}_{x,k}\Delta t, \end{array}$$
where *k* is the time instant, *x*, *y* are the coordinates of the vehicle in a fixed world frame, φ is the vehicle yaw, and *u* is the longitudinal velocity, Δt is the time interval spent in the current filter iteration. The state initialization is provided by the procedure described in Section 5.2.1. The controls fed to the model are taken from the INS measurements of the angular velocity $\hat{\omega }$ and the longitudinal acceleration ${\hat{a}}_{x}$. We chose this simple model for approximating the vehicle dynamics as the longitudinal acceleration is also a control input for the vehicle, and the yaw rate is directly measured by the INS. Other models such as point mass or single track can also be employed.

During the update phase, the measurements used are the velocity estimate coming from the OSS sensor and the orientation estimate coming from the Madgwick filter. Note that, as explained in Section 3, in absence of GNSS data, the INS velocity estimate is unreliable, which is why we use OSS instead. Moreover, this filter does not receive any absolute pose estimate, thus that part of the output will tend to diverge, while still providing a smooth velocity, which is what the subsequent filter needs to work correctly.

### 5.2. Lidar Scan Matching (IAMCL)

The goal of this filter is to give an absolute pose estimate of the vehicle relative to the offline built map using LiDAR data processing. The output of this filter is rather slow, as the input data has an update frequency of 25 Hz.

While the Adaptive Monte Carlo localization (AMCL) algorithm [40] has been widely employed for many practical applications, we encountered some limitations when dealing with our specific use case. In particular we experienced that:The filter automatic initialization provided in the AMCL ROS package [17] takes too much time to converge; in the racing context, however, the initialization must be accurate and should be performed before the car starts driving;During the race, many particles are generated outside of the racing track boundaries or with opposite orientation (with respect to the race fixed direction), which is inefficient;Due to unavoidable, even small, map imperfections (mainly false positives in the occupancy grid), the algorithm exhibits a kidnapping problem pretty often.

To tackle these issues we introduced two improvements: an automatic initialization procedure and a localization algorithm that injects a priori knowledge into the particles’ distribution.

#### 5.2.1. Automatic Initialization Procedure

We designed an automatic procedure to initialize all filters based on the initial LiDAR measurements and the offline built map. The idea is to generate a cloud of particles along a discretized racing track the center line, aligning the LiDAR scan with each particle and looking for the best match with the occupancy grid map. Particles are only drawn among the ones within the track borders and with orientation compatible with the race direction, which is known a priori. The weights are still computed in the same way as the classic Monte Carlo localization approach [41], using the likelihood field model. This procedure, described in Algorithm 1, returns an estimate of the initial pose of the vehicle, which is then used to initialize all the filters, as shown in Figure 6. The procedure is depicted in Figure 7.
**Algorithm 1:** Automatic initialization algorithm (*init*).
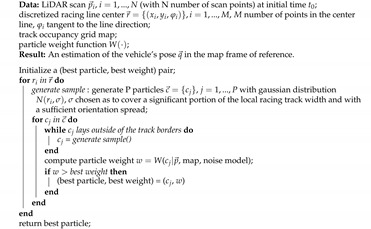


#### 5.2.2. Informed Prediction

The proposed particle filter is informed by sampling only in the positions allowed by the track, instead of sampling in the overall map space. A schematic view of the process is depicted in Figure 8.

A limit of a direct application of this idea is that, by constraining the particles to lie within the track borders, false positives are produced when the car happens to cross the track boundaries. This is not a concern for this application because, if this happened, a Race Control manager would call for an emergency stop. Nevertheless, we constantly monitor the weight of the particles and, in this case, if the value drops below a certain safety threshold, an emergency flag is raised. In this case, the control of the vehicle is handled by a separate emergency module in the real-time control board, which generates an emergency trajectory that allows the car to stop safely and avoiding wheel lock, even in the case of LiDAR failure, using as localization soures either GNSS (if available) or an Extended Kalman Filter based on velocity and acceleration measurements.

The theoretical framework of the particle filter for robot localization is described in [42]. A priori information can be injected into this framework during the *predict phase* by stating that
(2)$$\begin{array}{c} p\left(x_{k}|s_{k-1}^{i},u_{k-1}\right)=\left\{\begin{array}{cc} \tilde{p}\left(x_{k}|s_{k-1}^{i},u_{k-1}\right) & \mathrm{if}\;x_{k}\in I\\ 0 & \mathrm{if}\;x_{k}\notin I \end{array}\right. \end{array}$$
where *I* is the set of admissible robot states and $\tilde{p}\left(x_{k}|s_{k-1}^{i},u_{k-1}\right)$ is the probability of the robot state being $x_{k}$ at time *k* given the set of particles *S* at time k−1, and the controls u at time k−1. An explicit form of $\tilde{p}$ is not needed; instead, particles evolve according to the dynamic model when in an admissible state (see [41]), while they are redrawn around an external pose estimate when the state is not admissible, i.e., when the particle lies outside of the track boundaries. In the latter case, this pose estimate ${\hat{x}}_{k}$ is fed back from the smoothing filter described in Section 5.3, propagated forward in time using the model in Equation (1) to account for the asynchronicity among the filters:(3)$$\begin{array}{c} s_{k}^{\prime i}={\hat{x}}_{k}+\epsilon_{k}, \end{array}$$
where $s_{k}^{\prime i}$ is the newly generated *i*-th particle at time *k*, ${\hat{x}}_{k}$ is the latest smoothing filter pose estimate propagated at time *k* and $\epsilon_{k}$ is a gaussian noise. We choose $\hat{x}$ as output of the smoothing filter as it provides a robust localization estimate that is less affected by sudden pose jumps.

### 5.3. Smoothing Filter (EKF2)

The last element of the localization stack is an EKF that takes as input the OSS and the pose estimate from IAMCL. The goal of this filter is to provide a high frequency and smooth pose estimate to the vehicle control module that, in our experimental setup, runs on a real-time board. The filter is asynchronous: it iterates with a fixed frequency regardless of the sensor data received. New measurements from a sensor are stocked in a unit-length buffer, that is emptied as soon as they are used. This ensures a responsive output filter, the result of which is sent to the real-time board of the vehicle that computes the controls to the car. The system transition is based on the unicycle model (Equation (1)), where the initialization is provided by the procedure described in Section 5.2.1.

Due to the asynchronicity, with $N_{s}$ independent sources of measurement ($N_{s}$ = 2 in our case for OSS and pose estimate from IAMCL), there can be any of the $2^{N_{s}}$ combinations of measurements at each filter iteration, from no measurements at all, to all measurements concurrently available. The case with no measurements available involves the prediction step only. Each of these combinations requires a different observation model, defined as:(4)$$\begin{array}{c} z\left(k\right)=h\left(q(k)\right)+v\left(k\right) \end{array}$$
where $h:R^{m}\to R^{n}$ (*m* being the state dimensionality and *n* being the measurement dimensionality) is the observation model function, *z* the measurement vector, *v* the measurement noise, and
(5)$$\begin{array}{cc} q(k) & =\left[\begin{array}{c} x(k)\\ y(k)\\ \varphi (k)\\ u(k) \end{array}\right] \end{array}$$
is the overall filter state vector. In Table 1 we report the observation model jacobians (*H*) for some relevant cases.

This strategy is only applied to the output filter so that the localization module is always guaranteed to produce an updated pose estimate, regardless of communication delays and sensor faults, thus ensuring that the control module always receives a high frequency and smooth signal. Moreover, to improve driving safety, the EKF2 can also raise a flag where the pose estimate is insufficiently accurate. This is performed by evaluating the covariance returned from the Kalman Filter. The output of this filter is fed back to IAMCL, as shown in Figure 6, the use of which is described in Section 5.2.2.

## 6. Results

Experimental results were conducted in the Zala Zone (https://zalazone.hu/) proving ground, in the course of testing for the first Roborace localization challenge that took place in August 2019. For this event, regulations allowed mapping to be performed before the race with accurate GNSS, with the availability of the RTK system. On the other hand, during the race the GNSS system was manually degraded to make it unusable for localization. This allowed us to simulate a realistic scenario and focus on the issues arising in localization, due to sensor filtering and map distortion.

In the following, we perform several comparisons between different methods and in different testing conditions, both experimental and simulated. We analyze these tests in light of the error definitions reported in Figure 9, where position error with respect to a reference trajectory is computed. Let $\left(\hat{x},\hat{y}\right)$ be the estimated position of car, and (x,y) the reference position, i.e., the GNSS signal when available. Since all signals are synchronized, it is possible to calculate the Euclidean distance between estimated and reference positions. Given the resulting vector, it can be broken down into a lateral and a longitudinal component; the first one is the so called lateral error ($e_{LAT}$), and the second one is the longitudinal error ($e_{LON}$). Also, the heading error $e_{HEAD}$ is taken into account. These errors are computed as follows:(6)$$\begin{array}{cc} e_{\mathrm{LAT}}\left(k\right) & =|\left(\hat{x}-x\right)\sin\left(\psi \right)-\left(\hat{y}-y\right)\cos\left(\psi \right)|\\ e_{\mathrm{LON}}\left(k\right) & =\left(\hat{x}-x\right)\cos\left(\psi \right)+\left(\hat{y}-y\right)\sin\left(\psi \right)\\ e_{\mathrm{HEAD}}\left(k\right) & =\hat{\psi }-\psi \end{array}$$

The system was tested both in simulation and on a real racing vehicle, namely Roborace’s DevBot 2.0. Simulations were performed on the *rFpro* [43] simulator, with the identified vehicle model of the DevBot car provided by Roborace running on a 3D representation of the Monteblanco circuit (La Palma Del Condado, Spain) (https://www.circuitomonteblanco.com/). Experimental results were conducted in the Zala Zone proving ground (Zalaegerszeg, Hungary) in August 2019 during the Roborace localization challenge mentioned in Section 1. The track is shown in Figure 10. The track features very narrow corridors called *gates* (see Figure 11), only 0.5 m wider than the car, used to demonstrate the localization system effectiveness. The track consists of several cones that delimits the borders, a number of water barriers and a couple of inflatable obstacles mimicking pedestrians or other vehicles.

In the following, we first present an experimental comparison between our method and the original AMCL algorithm, followed by the analysis of the results on both a simulated and a real scenario.

### 6.1. Comparison With State-of-the-Art

To evaluate the effect of the proposed improvements, we compare the proposed localization method using the original AMCL implementation as a baseline. First, we remark that the initialization procedure described in Algorithm 1 is a novelty with respect to the baseline, hence no quantitative comparison can be made. We focus on comparing the whole localization stack described in Section 5 with the only difference in the LiDAR scan matching filter, while keeping every tuning parameter identical, including the ones of the EKF1 and EKF2 filters.

The comparison is performed on a dataset gathered during the trial week in the Zala Zone circuit, at a maximum speed of $v_{max}=60$ km/h. It is worth noting that, as the stack is run offline, the localization performance has no effect on the vehicle control. We compare two variants of IAMCL with AMCL: the first IAMCL variant corresponds to the algorithm described in Section 5.2 as it is, the second one is an improvement of that same algorithm we developed after the Zala Zone race. Indeed, during the race we observed a delay in the LiDAR scan matching module that mainly affected the longitudinal error, as it will be shown in Section 6.2. To tackle this problem, we measured the delay $\Delta t_{d}$ between the laser scan timestamp $t_{l}$ and the output pose generation and we extrapolated the actual pose at time $t_{l}+\Delta t_{d}$ with a forward Euler integration:(7)$$\begin{array}{cc} x(t_{l}+\Delta t_{d}) & =x\left(t_{l}\right)+v_{x}\left(t_{l}\right)\Delta t_{d}\\ y(t_{l}+\Delta t_{d}) & =y\left(t_{l}\right)+v_{y}\left(t_{l}\right)\Delta t_{d}\\ \theta (t_{l}+\Delta t_{d}) & =\theta \left(t_{l}\right)+\omega \left(t_{l}\right)\Delta t_{d}, \end{array}$$
where $\left(x\left(t_{l}\right),y\left(t_{l}\right),\theta \left(t_{l}\right)\right)$ is IAMCL output pose, and $\left(v_{x}\left(t_{l}\right),v_{y}\left(t_{l}\right),\omega \left(t_{l}\right)\right)$ are the Cartesian linear and angular velocities, as estimated from the odometry filter. In the following, we will refer to this improved version as IAMCL *with extrapolation*, whereas the original version (used during the tests) will be called IAMCL *without extrapolation*.

The dataset consists of several laps: here we report relevant results from the second and the beginning of the third lap, at the beginning of which we show an example of a situation where AMCL suffers from robot kidnapping and the proposed method does not. Results on the second lap are shown in Figure 12 where our method and AMCL achieve errors in the same order of magnitude. Relative to that figure, the average errors are reported in Table 2.

The localization errors obtained with the various methods indicate that: (i) extrapolation in IAMCL mitigates computational delays experienced during the tests, effectively bridging the gap with the baseline method in terms of Cartesian error.

This is mainly due to the particle distribution being denser in more meaningful regions of the state space, and the correction of the estimate prediction with (7). On the contrary, the heading error is always better for AMCL, and this might be due to over-constraining the resampled particles that violate track boundaries with a distribution chosen by the user around the latest EKF2 pose.

We can observe a sharp difference in the performance of the various methods after the beginning of the third lap. In this lap there is an increase of speed and acceleration that triggers a kidnapping failure for AMCL, with the consequent drastic increase of the error. The kidnapping occurrence is visible in Figure 13 where it is evident that while the baseline localization fails, the proposed method does not.

We noted that when the kidnapping problem arose with AMCL, the estimation of the covariance produced was unrealistic, which made it virtually impossible for the subsequent Extended Kalman Filter to handle this emergency situation. The most likely reason for the kidnapping to happen is because of a local poor map quality, which violates the LiDAR model assumed by the AMCL algorithm. Although this effect could likely be minimized with a better mapping algorithm tuning, it nevertheless represents a huge risk in the context of racing, as the map is often unknown until shortly before the race, which makes a more robust method preferable.

In conclusion, we choose to rely on our method, both because it shows overall better positional errors, but mostly because it is more robust to local poor scan matching.

### 6.2. Experimental Tests

We present the results of our localization system in three different cases:1.A run in autonomous mode at $v_{max}=60$ km/h;2.A run driven by a professional human driver at $v_{max}=100$ km/h, the maximum speed allowed by the track;3.A run in autonomous mode at $v_{max}=200$ km/h on the simulator.

As the results presented in this paper are based on data collected during preparation of an official event, not every dataset has GNSS data available to be used as a truth reference. In those datasets where GNSS is available (datasets 2 and 3), we compare the localization module outputs with GNSS data, and when that is not available (dataset 1), we compare it against the racing line, i.e., the desired reference pose.

In the following, we will show the results relative to the three datasets, using the variant of IAMCL with no extrapolation (see Section 6.1), as that improvement was developed only after the race. Future tests will include that improvement in experimental tests.

### 6.3. Dataset 1: Autonomous Mode (Experimental)

In Figure 14, the localization performances of the autonomous mode are shown. The control law is based on [44], and the maximum achieved speed is about 60 km/h for safety reasons. In this case the GNSS signal was cut off as a race rule, so no signal to use as a truth reference was available.

Because of this, the errors in Equation (6) were computed with respect to the desired trajectory (i.e., the racing line, reference for the controller module), instead of the GNSS signal. Thus, these errors include contributions not only from the localization system, but also from the control module (although indirectly). Reported metrics are the offtrack error (analogous to the lateral error in (Equation (6)) but with the racing line as reference) (max 0.17 m, mean 0.1 m) and the heading error (max 6.8 deg, mean 1.2 deg), both computed with respect to the racing line. Despite GNSS data being unavailable during this run, the car completed the lap successfully. As the reference trajectory is not temporized, it makes no sense to compute the longitudinal error in this case.

### 6.4. Dataset 2: Manual Drive Mode (Experimental)

As a second result, we report on experiments performed during a run driven by a professional pilot at the maximum velocity allowed by the track. Figure 15 shows the results relative to the fourth and fastest lap of this run ($v_{max}=100$ km/h). All errors are computed with respect to GNSS. The maximum lateral error is 0.64 m, with a mean of 0.18 m. The magnitude of the longitudinal error, about 10 times larger than the lateral (max 3.2 m, mean 1.9 m), is explained by the fact that the only absolute pose estimate provided to the output filter (EKF2) comes from IAMCL, which we measured having a response delay of 0.07 s. A car running at 100 km/h travels 2 meters in 0.07 s (a distance close to the observed longitudinal error), thus compatible with the time needed to complete an iteration of IAMCL. To compensate for this error, in future works we are going to use the IAMCL with the extrapolation improvement introduced in Section 6.1.

### 6.5. Dataset 3: Autonomous Mode (Simulation)

Finally, in Figure 16, localization performances of a simulation run in the Monteblanco circuit are reported. Maximum longitudinal velocity was 200 km/h, performed on the Roborace simulation system. In this case, the longitudinal error (max 8.4 m, mean 4.92 m) is even larger than in the previous case, nevertheless its magnitude is compatible with the measured IAMCL delay, as described in Section 6.4. The maximum lateral error is 1.48 m, with a mean of 0.25 m, the maximum heading error is 3.52 degrees, with a mean of 1.6 degrees. These results show the effectiveness of the proposed system also at high speeds, demonstrated by the magnitude of the lateral error that has maintained a reasonable level. The longitudinal error shows instead a significative increase, which confirms its dependency from the vehicle speed but that can be limited by using the IAMCL with extrapolation improvement described in Section 6.1, as for Dataset 2.

## 7. Conclusions

This paper tackles the problem of localization in a GNSS-denied environment with a LiDAR-based system. The proposed method relies on two EKFs and a particle filter for LiDAR scan matching, the latter exploiting a priori information about the environment to build the particle set in a more efficient way. The envisioned application is that of autonomous racing vehicles. Reported results show good performance both in open loop (localization performed online during manual driving) and closed loop (with the pose estimate sent in feedback to the controller). Notably, our method shows robustness against the kidnapped robot failure with respect to a widely used state-of-the-art AMCL implementation. Future works will be devoted to further optimization by means of implementation in Compute Unified Device Architecture (CUDA) of the particle filter, to better exploit the GPU-based hardware available. Moreover, we aim at testing the system in a multi-vehicle racing scenario and at higher speeds.

## Figures and Tables

**Figure 1 sensors-20-03992-f001:**
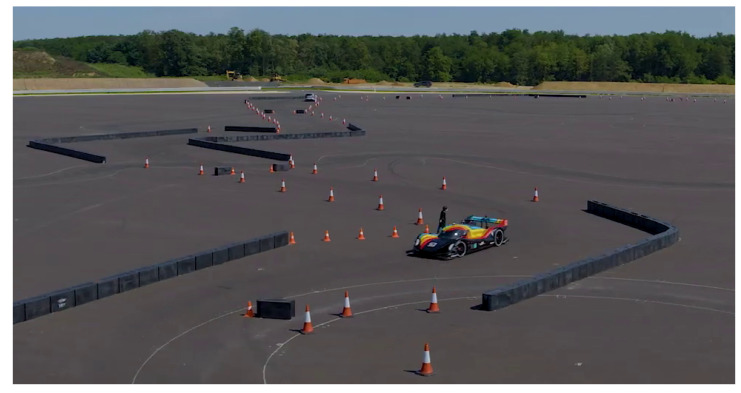
Roborace DevBot 2.0 in the Zala Zone Circuit. Localization challenges were performed in this circuit, and the presented localization framework was used.

**Figure 2 sensors-20-03992-f002:**
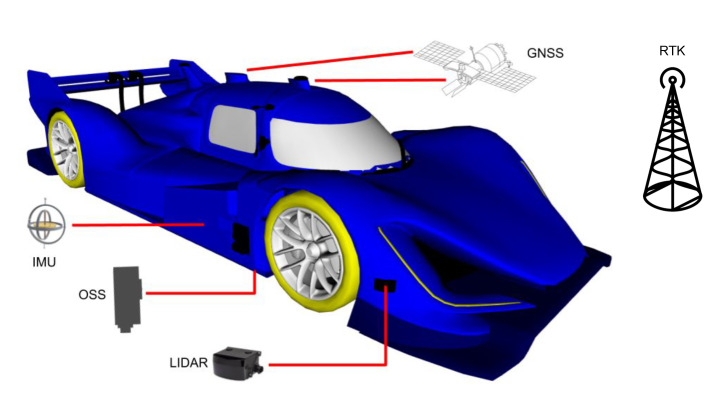
Overview of the DevBot sensors. Light Detection and Ranging (LiDARs) are mounted in the front, side and back of the car. Global Navigation Satellite Systems (GNSS) and Inertial Measurement Unit (IMU) sensors come from the OxTS system and the Optical Speed Sensor (OSS) measures longitudinal and lateral car velocities. The Real-Time Kinematic (RTK) base station is an optional system that allows extremely high positioning precision to the OxTS system.

**Figure 3 sensors-20-03992-f003:**
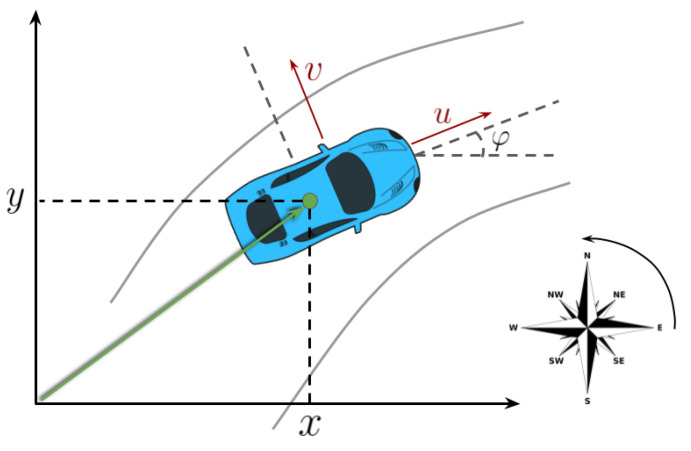
The adopted sensor measurement reference frame. *u* and *v* are the longitudinal and lateral velocities; *x* and *y* are the vehicle coordinates with respect to the map origin.

**Figure 4 sensors-20-03992-f004:**
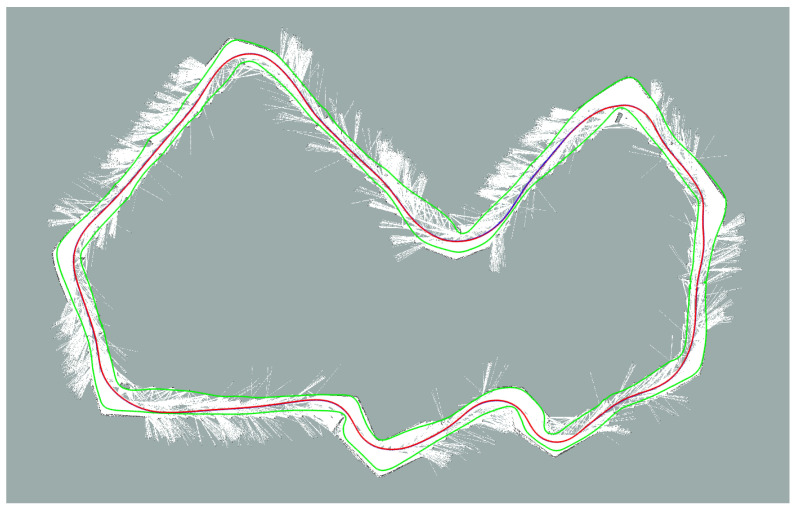
The map built for the race. The circuit track borders are represented by the two green lines, white areas are the obstacle free spaces, while the red line represents the racing line, to be followed during the run.

**Figure 5 sensors-20-03992-f005:**
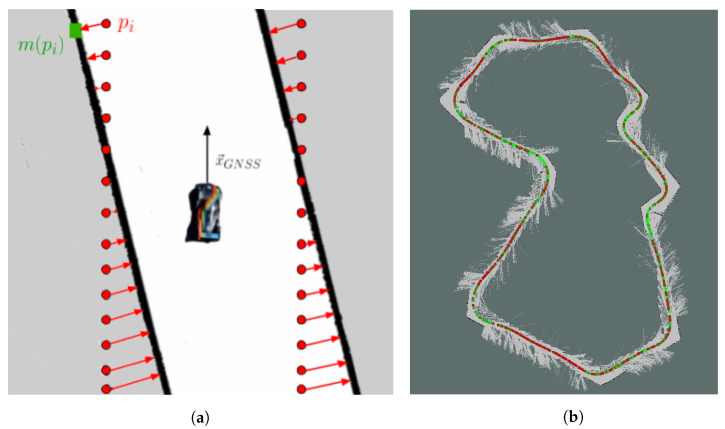
(**a**) Computation of the mapping error $e_{map}\left(\xi^{*}\left(k\right)\right)$. Given a LiDAR reading place on a GNSS pose, each laser scan point ($p_{i}$ in red) is compared with its closest map occupied point ($m_{p_{i}}$ in green); (**b**) Result of the map quality assessment procedure on the Zala Zone track map. The circular markers are placed on GNSS positions; the colors, from green to red, represent the values of $e_{map}$ (green markers correspond to lower values of $e_{map}$).

**Figure 6 sensors-20-03992-f006:**
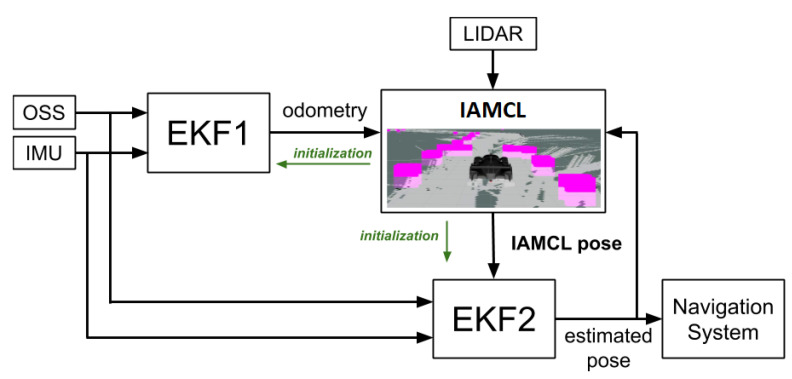
Localization pipeline overview. OSS and IMU data feed the Extended Kalman Filters (EKFs); EKF1 produces an odometry estimate which is sent to the Informed Adaptive Monte Carlo Localization (IAMCL) module. By comparing LiDAR data with the pre-built map, IAMCL outputs a pose estimate that is then sent to EKF2, a smoothing filter with a high-frequency output. The green arrows indicate the initialization procedure described in Section 5.2.1.

**Figure 7 sensors-20-03992-f007:**
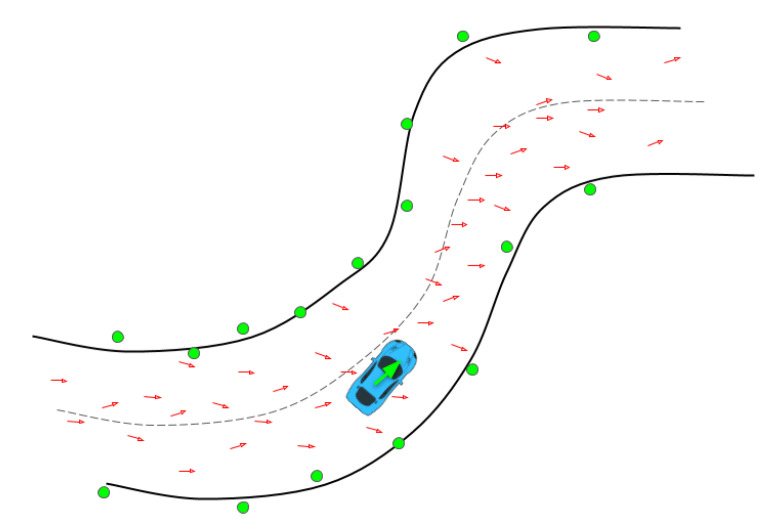
Representation of the automatic initialization procedure: a number of particles is generated with a Gaussian distribution around the track center line within the admissible state space (the inner area of the racing track), while limiting the particles orientation along the race direction. Each particle is represented by a red arrow; we rotate and translate the LiDAR point cloud around each red arrow, and compute the best matching particle, whose resulting match is shown in green.

**Figure 8 sensors-20-03992-f008:**
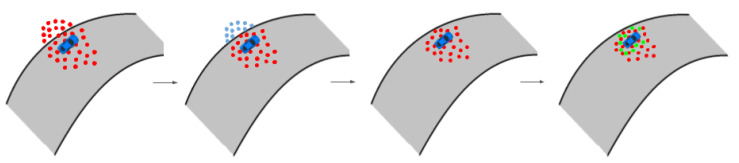
Illustrative schema of the IAMCL algorithm. The particles are first drawn with a distribution around an initial pose, then the ones with states that violate the track boundaries are eliminated and redrawn in the admissible state space.

**Figure 9 sensors-20-03992-f009:**
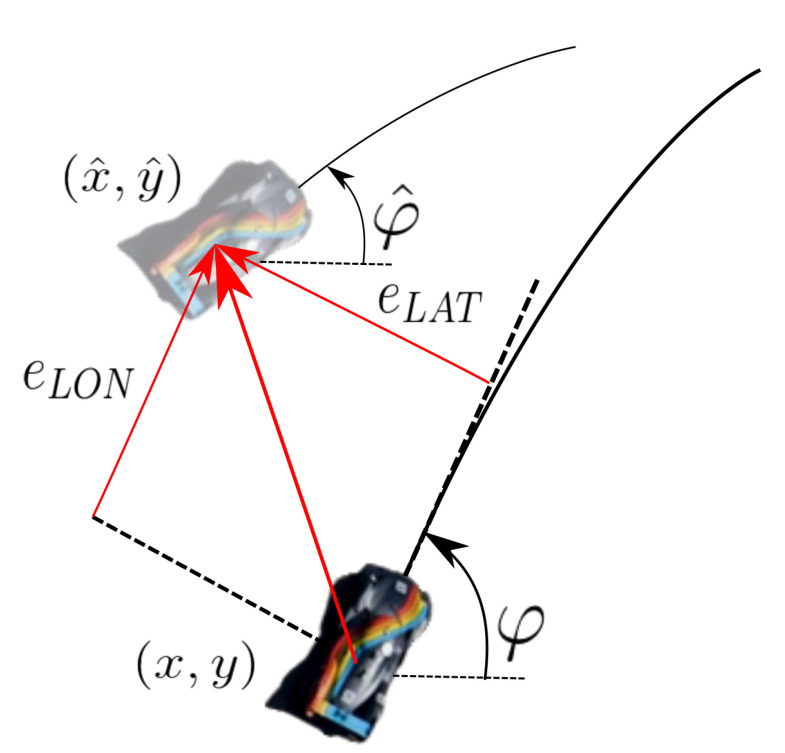
Longitudinal and lateral errors. These are computed as the distance error components along the GNSS defined car orientation direction.

**Figure 10 sensors-20-03992-f010:**
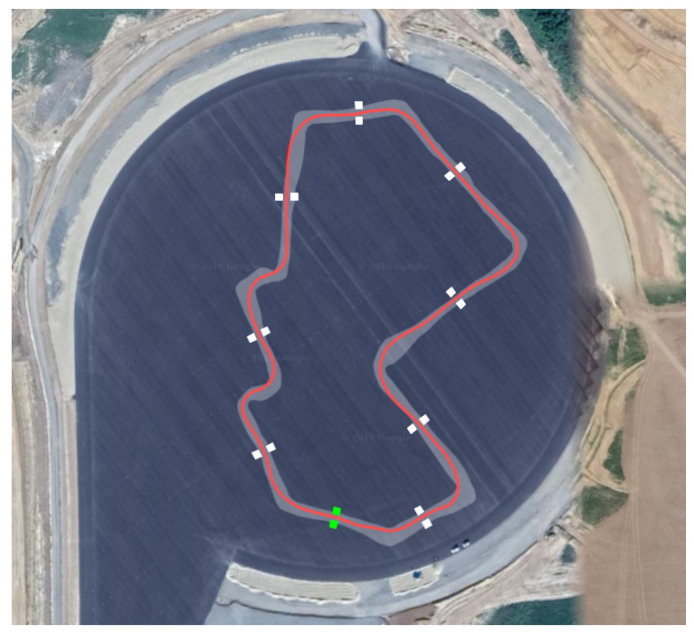
Satellite view of Zala Zone proving ground. Track boundaries and the racing line are shown in red. Gates are shown in white and the starting point in green.

**Figure 11 sensors-20-03992-f011:**
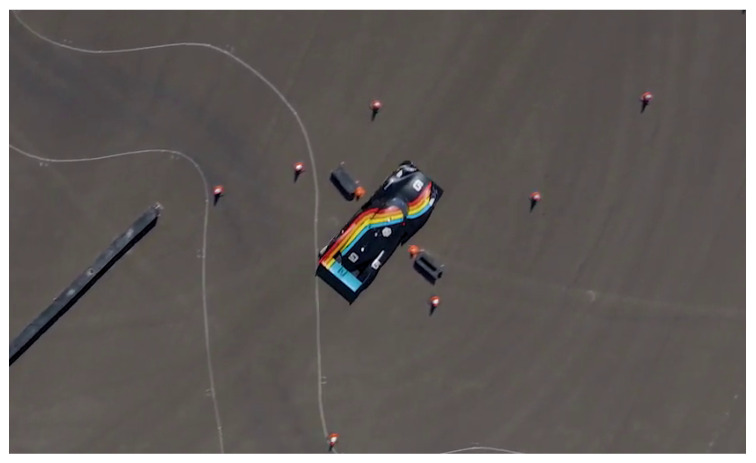
Top view of the Roborace DevBot 2.0 in Zala Zone Circuit. Narrow corridors (gates) were distributed along the track, the car is 2 meters wide, while the gates were 2.5 meters wide.

**Figure 12 sensors-20-03992-f012:**
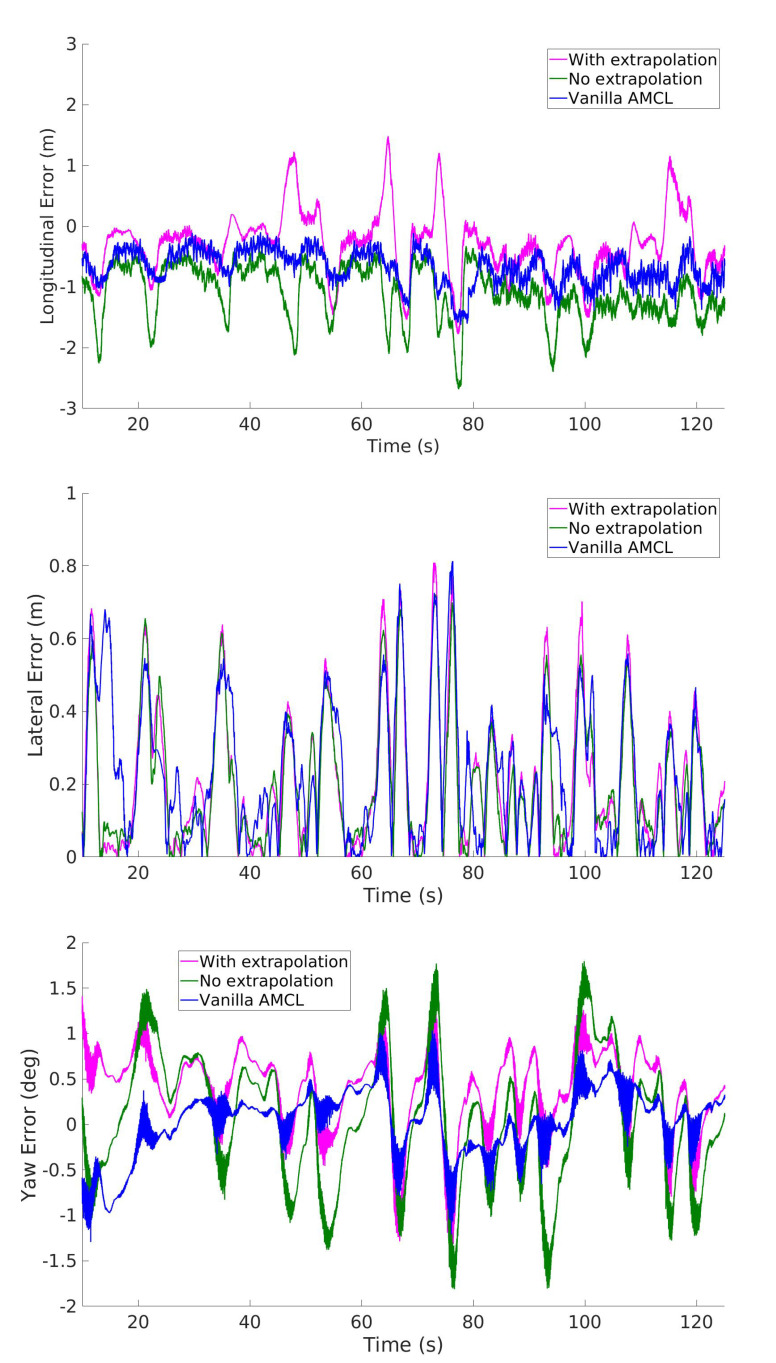
Comparison of the longitudinal, lateral and heading errors of two variants of IAMCL (with and without the extrapolation defined in Equation (7)) and the Adaptive Monte Carlo Localization (AMCL) algorithm during the second lap of the dataset.

**Figure 13 sensors-20-03992-f013:**
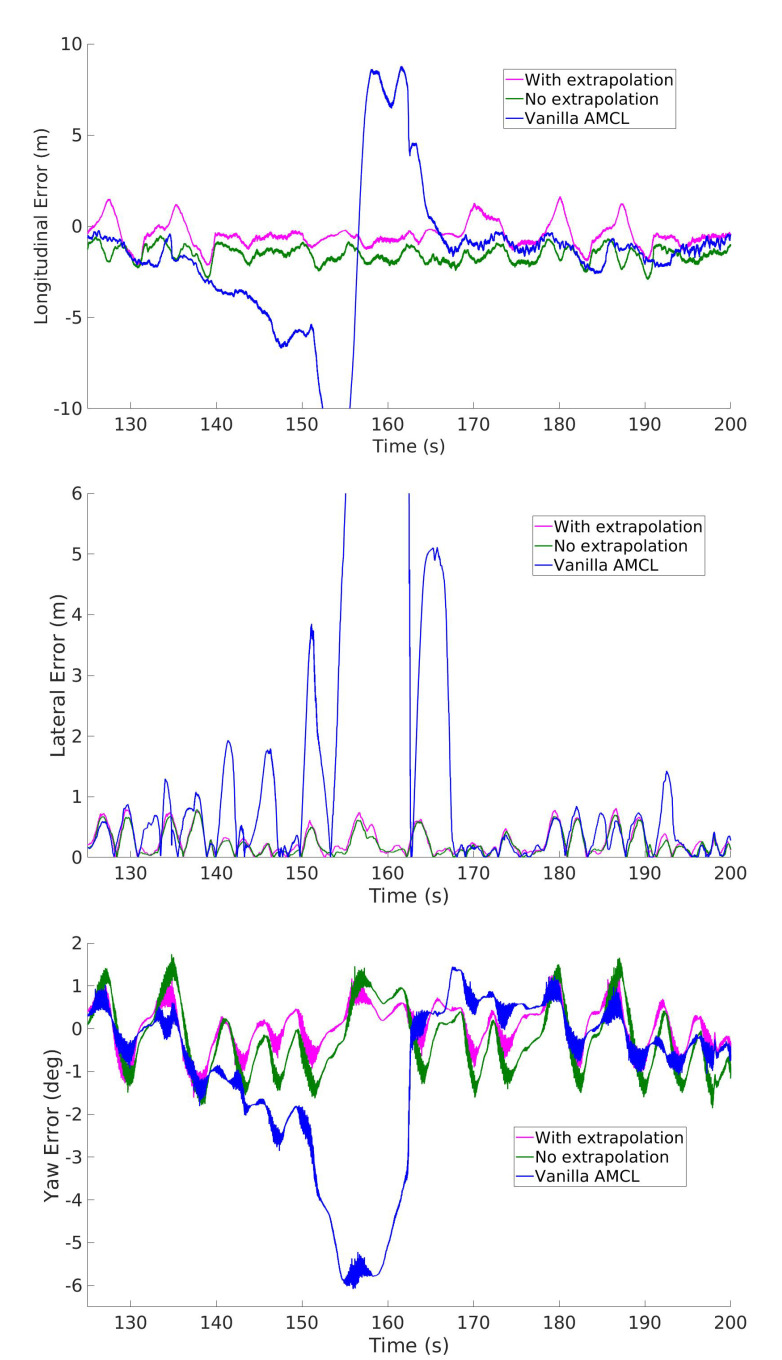
Comparison of the longitudinal, lateral and heading errors of two variants of IAMCL (with and without the extrapolation defined in Equation (7)) and AMCL during the third lap of the dataset, with focus on an AMCL failure (due to robot kidnapping). Note that IAMCL does not suffer from such situation even if they share the same underlying scan matching algorithm.

**Figure 14 sensors-20-03992-f014:**
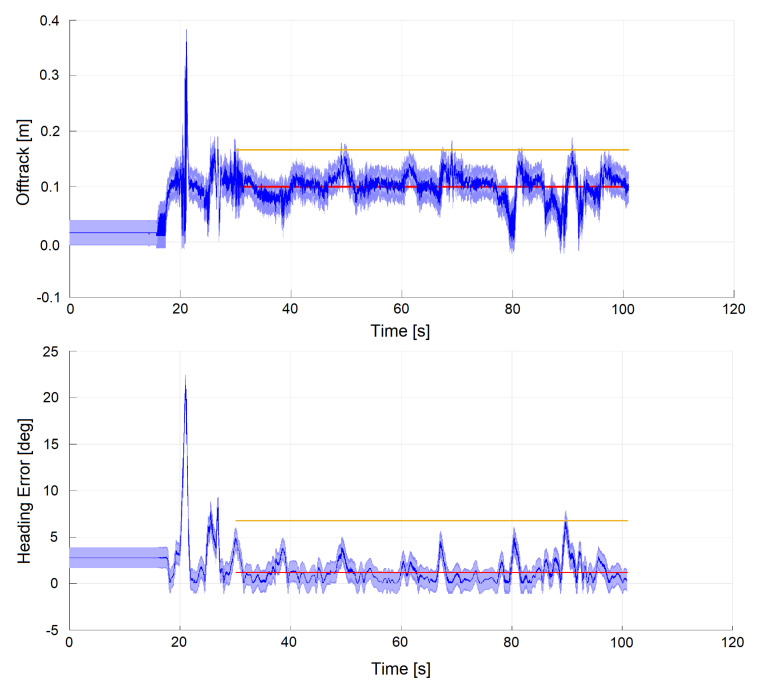
Autonomous lap at $v_{max}=60$ km/h. Signals are plotted together with standard deviation (light blue), mean (red) and maximum error (yellow). The measurement of such metrics starts when the car actually starts driving after initialization.

**Figure 15 sensors-20-03992-f015:**
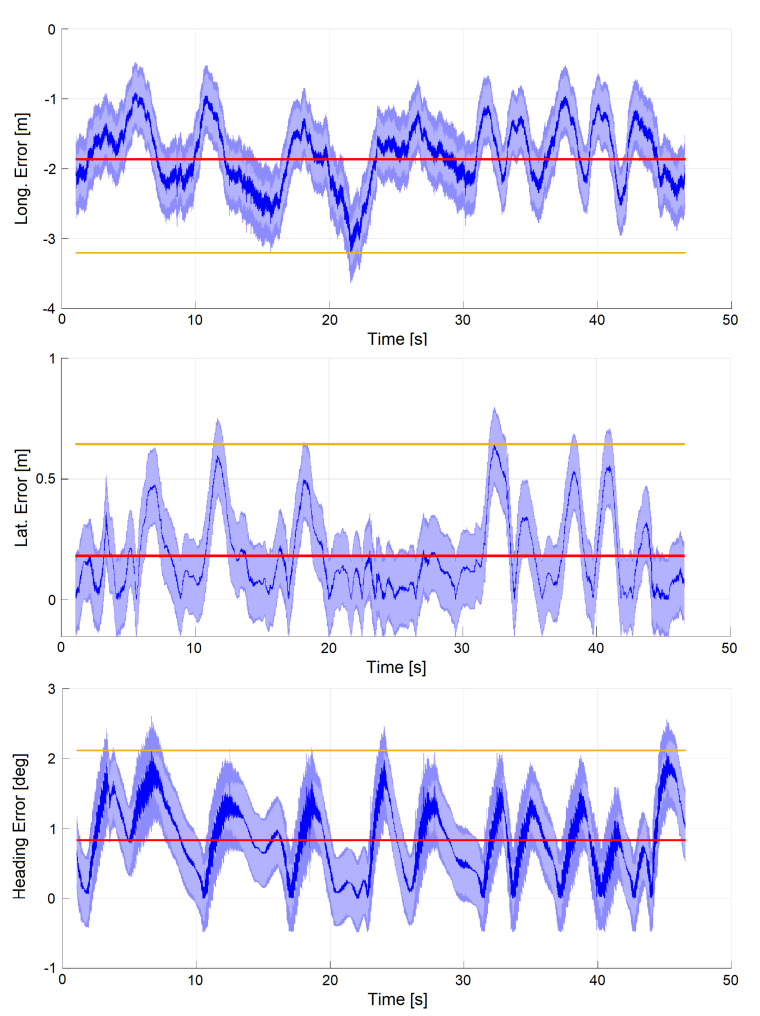
Manually driven lap at $v_{max}=100$ km/h. Signals are plotted together with standard deviation (light blue), mean (red) and maximum error (yellow).

**Figure 16 sensors-20-03992-f016:**
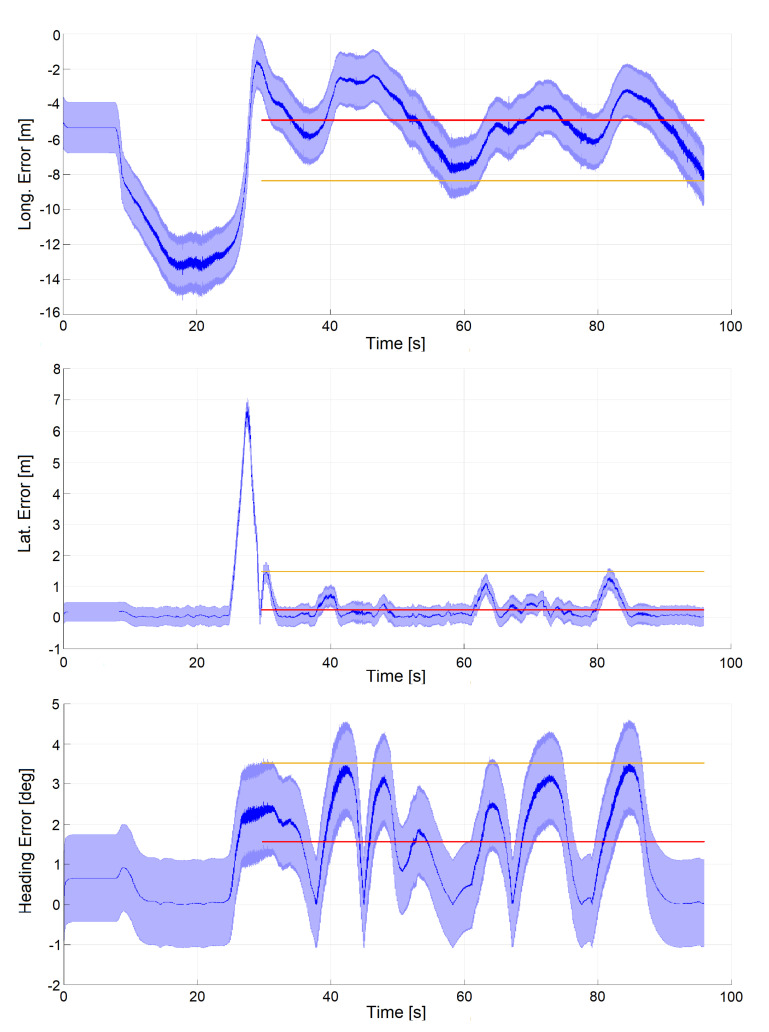
Autonomous lap in simulation at $v_{max}=200$ km/h. Signals are plotted together with standard deviation (light blue), mean (red) and maximum error (yellow). The measurement of such metrics starts when the car actually starts driving after initialization.

**Table 1 sensors-20-03992-t001:** Some possible observation model Jacobians.

Case	*H*	h(q(k))
velocity not available	$\left[\begin{array}{cccc} 1 & 0 & 0 & 0\\ 0 & 1 & 0 & 0\\ 0 & 0 & 1 & 0 \end{array}\right]$	$\left[\begin{array}{c} x\\ y\\ \varphi \end{array}\right]$
pose estimate not available	$\left[\begin{array}{cccc} 0 & 0 & 0 & 1 \end{array}\right]$	$\left[\begin{array}{c} u \end{array}\right]$
all measurements available	$I_{4x4}$	$\left[\begin{array}{c} x\\ y\\ \varphi \\ u \end{array}\right]$

**Table 2 sensors-20-03992-t002:** Comparison between the average longitudinal, lateral and heading average errors of the localization stack using IAMCL (with and without extrapolation) or AMCL.

	IAMCL (w/ extr.)	IAMCL (w/o extr.)	AMCL
**Long. error (avg/max)**	0.47/1.78 m	1.10/2.69 m	0.68/1.63 m
**Lat. error (avg/max)**	0.21/0.81 m	0.20/0.72 m	0.23/0.81 m
**Heading error (avg/max)**	0.51/1.39 deg	0.57/1.81 deg	0.29/1.29 deg

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
