# Peer review of "LiDAR-Based GNSS Denied Localization for Autonomous Racing Cars"

_sensors, 2020, doi:10.3390/s20143992_

Round 1

Reviewer 1 Report

The authors introduce a location architecture with a method based on particle filtering consisting of two multi-rate Kalman Filters and on a LiDAR processing algorithm named Informed Adaptive Monte Carlo localization. In my opinion, the topic is of interest to the readers, and the method and algorithm are adequately described. However, the paper would be reconsidered after major revision for the following suggestions:

  1. In general, there is lack of explanation of the problem discussed in the paper. Moreover, the purpose and results of the research are vague. I think the authors should set the problem clearer and write one section to define the problem.
  2. In Abstract, transform the status quo into the problem to be solved and the relevant solutions to it. And more focuses on quantitative analysis of experimental results are needed.
  3. In section 1, as filter-based approaches are adopted in the study, current research and improvement regarding to filtering should discussed. And I think it is better to move description part about race in respect of into section 5.
  4. In section 2, it’s better to add the specific graphical representation of the mentioned reference frame and motion parameters make sense.
  5. In section 3.1, with accurate GNSS, the localization problem is solved. Is it possible to generated a better map with GNSS data through the OpenSlam’s Gmapping?
  6. In section 4. here, the claimed ‘we describe the architecture of the four filters’ is confusing, because the architecture has three main filters (EKF1, EKF2 and IAMCL)
  7. In section 4.1, the authors did not state clearly why OSS sensor were chosen rather than INS sensor. Compared with INS sensor, what’s advantages of the OSS sensor?
  8. In section 4.2.1, it’s not clear how the particles distribute, when a cloud of particle was generated.
  9. In section 5, there is a lack of comparison of the proposed methods between the existing methods or popular methods, as well as a lack of quantitative analysis. As a consequent, the proposed location architecture is little convincing for readers.
  10. Through the full paper, some mistakes of figure arrangement. Figure 1, 5, 6 should be pasted after the mentioned paragraph. And large white space should be avoided by reconsider typography and charts. Figure 9, 10 and 11 should Maintain a consistent style in size.

Reviewer 2 Report

Overview:

The authors of the paper propose a localization framework for autonomous cars racing on a GNSS-denied racing track. In this paper, the authors use odometry and LiDAR data in order to perform localization based on a map that is captured offline. The authors develop a framework that consists of two Kalman filters and a LiDAR based particle filter. While the first Kalman filter uses odometry information to perform high frequency pose estimates, the particle filter provides global estimates using the low frequency LiDAR data. Finally, the estimates from these two modules are combined by another extended Kalman filter to get a smoothed high frequency pose estimate. For the LiDAR particle filters, an automatic initialization procedure is used and the kidnapping problem is handled by injecting a-priori knowledge. The automatic initialization involves matching the LiDAR scan with the occupancy grid of the map to find the best match, initializing all filters, and improving accuracy. A-priori knowledge about the valid locations on the map, on the other hand, helps to reduce the kidnapping problem.

Strengths:

  1. The paper develops a framework for solving the localization problem for self driving racing cars using particle and Extended Kalman Filters by using a pre-race map.
  2. Combines information from multiple sensors to get frequent and smooth pose estimates in an open loop or a closed loop application.
  3. Introduces the use of automatic initialization and a-priori knowledge to improve the localization accuracy under the constraints given by the specific race considered in the paper. 
  4. Attempts to handle the localization problem in a multi-rate environment with the need to achieve accurate, high frequency estimates.

Weaknesses:

  1. The a-priori information provided reduced the kidnapping problem but it assumes that the Race Control Manager is expected to stop the car if it leaves the race track. A backup plan is provided to stop the car if the particle weights change significantly. However, in a real race with multiple vehicles, a car that halts suddenly while it is partially out of the race track can cause a hazardous situation. In this specific case, there is only one vehicle on the track at a time, so the only concern is about the vehicle being damaged if it moves out of course.
  2. While this is a hard problem, and it is understandable why the car was not tested at higher speeds in the autonomous mode, testing it at 60 km/hr in autonomous mode does not prove that the localization framework can be used in a production system for an actual (human) car race.
  3. The results indicate a large longitudinal error and while the reason is discussed, the high error value is not desirable.
  4. The error increases as the speed of the vehicle increases. This may be inevitable but requires scrutiny because the aim is to perform localization of high speed autonomous vehicles.
  5. No empirical comparison is made to existing techniques for localization.

Questions:

  1. Since this car has actually operated on the race course discussed in the paper, can you provide some details about how weakness point 1 was handled during the actual race? Is there another part of the system that is responsible for handling this?
  2. When the map is generated before the race, are GNSS, odometry and LiDAR data used and captured? Specifically are you saving any LiDAR data during the mapping phase and using it for matching later? Or are you using the LiDAR and odometry as well as GNSS to generate an occupancy grid and matching the LiDAR data to the occupancy grid during the race?
  3. The localization technique proposed here is the one that was actually used during the race or is this an improved one that was not actually used to control the car during the race?

Accept after major revision. 

  1. Minimally, the work must be compared experimentally to one other existing localization technique even if that is your own previous localization technique. If it is not possible to do so, concrete reasons must be provided for that. 
  2. Experimental comparison to one filter-based method and one deep-learning based method is preferable.
  3. Introduction should include more information on related work.
  4. Please add references to refer to existing deep learning techniques for localization and compare your technique conceptually to other existing localization/ SLAM techniques. Some papers you may consider are provided below. All of these solve the SLAM problem in different scenarios.
  1. https://arxiv.org/abs/1612.00380
  2. https://www.sciencedirect.com/science/article/abs/pii/S0921889018308029
  1. Please answer the questions I have provided under the questions section.

Round 2

Reviewer 1 Report

I’m glad to re-review the paper in greater depth. The authors have substantially revised the paper and addressed most of the comments. I think the paper would be accepted after minor revision:

Figure 14, 15 and 15 should maintain a consistent style in size and annotation. And avoid too much empty space, for instance in page 20.

Author Response

We are glad that the reviewer was satisfied of this revision of the paper. We have fixed the Fig. 14, 15 and 16's sizes and captions so that they are consistent, and moved Fig. 15 and 16 so that there is no white space.

The only change to the text is in the caption of figure 16, which became:

"Autonomous lap in simulation at vmax=200km/h..." instead of "Simulation results, at vmax=200km/h..." to make it consistent with the other two figures.

We would like to thank the reviewer for the valuable comments that led to the significant improvement of our work.

Reviewer 2 Report

The concerns raised in the first review have been adequately addressed.

  1. The introduction now has adequate references and discussion about related work.
  2. Comparison has been made empirically to an existing localization method.
  3. Adequate reason provided for not comparing to deep learning based methods.
  4. Made an improvement to the previous technique to increase the accuracy and address some concerns.

Author Response

We are glad the reviewer was satisfied of the last revision and we thank him for the valuable comments that contributed to this significant improvement.